# Metabolomic profiling of microbial disease etiology in community-acquired pneumonia

Ilona den Hartog[1], Laura B. Zwep[1,2,3], Stefan M. T. Vestjens[4], Amy C. Harms[1], G. Paul Voorn[4], Dylan W. de Lange[5,6], Willem J. W. Bos[7,8], Thomas Hankemeier[1], Ewoudt M. W. van de Garde[9,10], J. G. Coen van Hasselt[1]*

**1** Division of Systems Biomedicine & Pharmacology, Leiden Academic Centre for Drug Research, Leiden University, Leiden, The Netherlands, **2** Mathematical Institute, Leiden University, Leiden, The Netherlands, **3** Leiden Centre of Data Science, Leiden University, Leiden, The Netherlands, **4** Department of Medical Microbiology and Immunology, St. Antonius Hospital, Nieuwegein, The Netherlands, **5** Intensive Care, University Medical Center Utrecht, Utrecht University, Utrecht, The Netherlands, **6** National Poison Information Center, University Medical Center Utrecht, Utrecht University, Utrecht, The Netherlands, **7** Department of Internal Medicine, St. Antonius Hospital, Nieuwegein, The Netherlands, **8** Department of Internal Medicine, Leiden University Medical Center, Leiden, The Netherlands, **9** Division of Pharmacoepidemiology and Clinical Pharmacology, Utrecht University, Utrecht, The Netherlands, **10** Department of Clinical Pharmacy, St. Antonius Hospital, Nieuwegein, The Netherlands

* coen.vanhasselt@lacdr.leidenuniv.nl

**Data Availability Statement:** All relevant data are within the manuscript and its Supporting Information files.

## Abstract

Diagnosis of microbial disease etiology in community-acquired pneumonia (CAP) remains challenging. We undertook a large-scale metabolomics study of serum samples in hospitalized CAP patients to determine if host-response associated metabolites can enable diagnosis of microbial etiology, with a specific focus on discrimination between the major CAP pathogen groups *S. pneumoniae*, atypical bacteria, and respiratory viruses. Targeted metabolomic profiling of serum samples was performed for three groups of hospitalized CAP patients with confirmed microbial etiologies: *S. pneumoniae* (n = 48), atypical bacteria (n = 47), or viral infections (n = 30). A wide range of 347 metabolites was targeted, including amines, acylcarnitines, organic acids, and lipids. Single discriminating metabolites were selected using Student's T-test and their predictive performance was analyzed using logistic regression. Elastic net regression models were employed to discover metabolite signatures with predictive value for discrimination between pathogen groups. Metabolites to discriminate *S. pneumoniae* or viral pathogens from the other groups showed poor predictive capability, whereas discrimination of atypical pathogens from the other groups was found to be possible. Classification of atypical pathogens using elastic net regression models was associated with a predictive performance of 61% sensitivity, 86% specificity, and an AUC of 0.81. Targeted profiling of the host metabolic response revealed metabolites that can support diagnosis of microbial etiology in CAP patients with atypical bacterial pathogens compared to patients with *S. pneumoniae* or viral infections.

**Funding:** This work is part of the research program 'Metabolomic fingerprint biomarkers to guide antibiotic therapy and reduce resistance' with project number 541001007, which is financed by ZonMW, the Netherlands Organization for Health Research and Development associated with the Dutch Research Council (NWO). The funders had no role in study design, data collection and analysis, decision to publish, or preparation of the manuscript.

**Competing interests:** The authors have declared that no competing interests exist.

## Introduction

Community-acquired pneumonia (CAP) is a commonly occurring respiratory tract infection caused by bacterial or viral pathogens that can lead to severe disease, especially in elderly patients [1]. The predominant pathogens found in hospitalized CAP patients are *Streptococcus pneumoniae* and to a lesser extent, *Haemophilus influenzae*, *Legionella pneumophila*, and respiratory viruses [2, 3]. Patients hospitalized with severe CAP typically receive empirical antibiotic treatment with broad-spectrum antibiotics until the microbial etiology is determined [4, 5]. Current standard diagnostic methods for microbial identification are pathogen-targeted and include culturing, antigen testing, and molecular diagnostics such as PCR [5]. In over 60% of CAP patients, no causative pathogen can be identified with these pathogen-targeted diagnostic techniques [2, 6]. As a consequence, broad-spectrum antibiotics are over-used, which facilitates the emergence of antimicrobial resistance [7, 8]. To this end, a need exists to explore innovative methods to enhance the diagnostic performance for the detection of microbial pathogens in CAP.

Evaluation of differences in the host-response to CAP-associated pathogens may be an alternative approach to improve diagnosis [9]. There is growing evidence that the host, i.e. the patient, metabolic response to infections can be a relevant source of novel host immune response biomarkers to infections [10, 11]. Several small studies have reported differences in metabolite profiles in blood and urine samples in patients with different types of infections (S1 Table) [12–18]. For instance, studies comparing metabolomic changes in CAP and tuberculosis (TB) patients show increased levels of plasma lipids and decreased levels of metabolites involved in cholesterol synthesis [12, 15]. A study comparing viral and bacterial respiratory tract infections showed that plasma metabolite profiles of patients with influenza A and bacterial pneumonia differed significantly [17]. In another study, urine samples of patients with a respiratory syncytial virus (RSV) or a bacterial respiratory tract infection showed differences in metabolite levels as well [18]. An important limitation of these studies is that the comparisons made cannot yet support the etiological diagnosis of CAP but merely focus on differences between diseases such as TB versus CAP. The studies that compared viral and bacterial causative pathogen groups of CAP used an untargeted metabolomics approach. While an untargeted approach is especially useful for the discovery of new features and hypothesis-free analysis, a targeted approach that can be fully quantified to clinical laboratory standards may be preferable for clinical implementation. Furthermore, these studies have the limitation that they focus on the comparison of pediatric patients while most hospitalized CAP patients are adults. No studies have evaluated differences in metabolite profiles of CAP patients comparing different microbial etiologies relevant for treatment of CAP, i.e. *S. pneumoniae*, atypical pathogens, and viral infections.

In the current study, we performed extensive targeted metabolomic profiling for three groups of hospitalized CAP patients with confirmed microbial etiologies of *S. pneumoniae*, atypical bacteria, or viral infections. We aimed to determine whether host-response associated metabolites can enable diagnosis of microbial etiology, focusing on discrimination between the pathogen groups *S. pneumoniae*, atypical bacteria, and respiratory viruses in patients hospitalized with CAP.

## Materials and methods

### Study population

Serum samples were taken from 505 patients that were diagnosed with CAP in two previously conducted clinical studies that were executed between October 2004 and September 2010 [2,

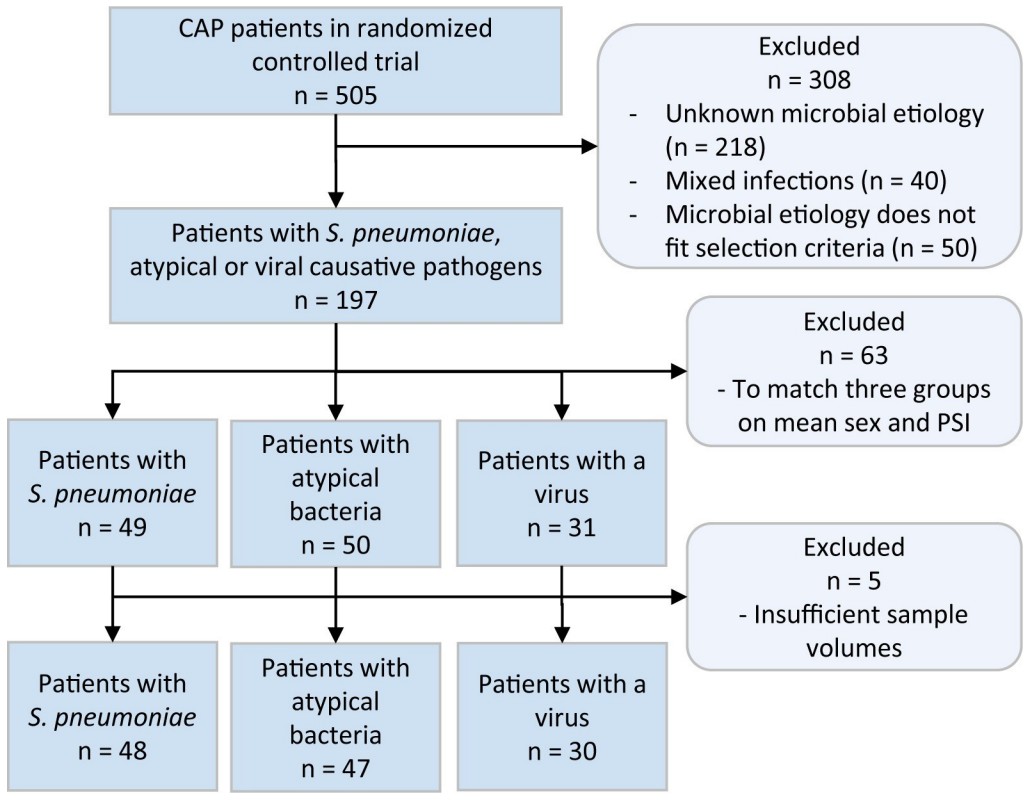

**Fig 1. Flow chart of the formation of the three studied patient groups.**

3]. The samples were taken from CAP patients within 24 hours after hospital admission. In 57% of these patient samples, the causative pathogen could be identified using conventional diagnostic methods such as culturing, PCR, and urinary antigen tests. The most commonly found causative pathogen in these patients was *S. pneumoniae*, followed by atypical bacterial and viral pathogens. A minority of patients was diagnosed with other bacteria.

From the selection of patients in which a causative pathogen was identified, we excluded patients with mixed infections. Furthermore, we constructed three distinctive groups of patients with *Streptococcus pneumoniae*, atypical (*Coxiella burnetii*, *Chlamydophila psittaci*, *Legionella pneumophila* or *Mycoplasma pneumoniae*), or viral (influenza virus, herpes simplex virus (HSV), respiratory syncytial virus (RSV), parainfluenza virus, or another respiratory virus) infections. The number of available samples for the patient group with confirmed viral CAP infection was limited (n = 31). The patients included in the *S. pneumoniae* and atypical bacterial groups were randomly drawn from the remaining study population in an iterating fashion until the bacterial groups were composed in such a way that three groups showed comparable means for sex and pneumonia severity index scores. This resulted in a group of 49 patients with *S. pneumoniae* and a group of 50 patients with atypical infections (Fig 1). No matching of individual samples was performed. An overview of patient characteristics is provided in Table 1 and S2 Table. Patient characteristics that might be considered as possible covariates were: age, sex, nursing home resident, renal disease, congestive heart failure, CNS disease, malignancy, COPD, diabetes, altered mental status, respiratory rate, systolic blood pressure, temperature, pulse, pH, BUN, sodium, glucose, hematocrit, partial pressure of oxygen, pleural effusion on x-ray, duration of symptoms before admission, antibiotic treatment

**Table 1. Patient characteristics per pathogen group.**

| | *S. pneumonia* (n = 48) | Atypical (n = 47) | Viral (n = 30) | P-value |
|---|---|---|---|---|
| **Age (years)** | | | | |
| Mean (SD) | 62.2 (18.9) | 54.7 (14.6) | 70.1 (16.4) | <0.01 |
| Median [Min, Max] | 63.5 [18.0, 98.0] | 52.0 [26.0, 81.0] | 74.0 [29.0, 95.0] | |
| **Sex** | | | | |
| Male | 22 (45.8%) | 34 (72.3%) | 21 (70.0%) | 0.12 |
| **PSI score** | | | | |
| < 50 | 9 (18.8%) | 9 (19.1%) | 2 (6.7%) | 0.33 |
| 51–70 | 7 (14.6%) | 13 (27.7%) | 6 (20.0%) | |
| 71–90 | 5 (10.4%) | 10 (21.3%) | 7 (23.3%) | |
| 91–130 | 23 (47.9%) | 12 (25.5%) | 11 (36.7%) | |
| > 131 | 4 (8.3%) | 3 (6.4%) | 4 (13.3%) | |
| **Liver disease** | | | | |
| No | 48 (100%) | 47 (100%) | 30 (100%) | - |
| **Kidney disease** | | | | |
| Yes | 3 (6.2%) | 1 (2.1%) | 4 (13.3%) | 0.30 |
| **Cardiovascular disease** | | | | |
| Yes | 6 (12.5%) | 5 (10.6%) | 3 (10.0%) | 0.93 |
| **CNS disease** | | | | |
| No | 46 (95.8%) | 44 (93.6%) | 28 (93.3%) | 0.66 |
| Yes | 1 (2.1%) | 3 (6.4%) | 2 (6.7%) | |
| Missing | 1 (2.1%) | 0 (0%) | 0 (0%) | |
| **Malignancy** | | | | |
| No | 44 (91.7%) | 46 (97.9%) | 28 (93.3%) | 0.66 |
| Yes | 3 (6.2%) | 1 (2.1%) | 2 (6.7%) | |
| Missing | 1 (2.1%) | 0 (0%) | 0 (0%) | |
| **COPD** | | | | |
| No | 24 (50.0%) | 44 (93.6%) | 25 (83.3%) | 0.16 |
| Yes | 9 (18.8%) | 3 (6.4%) | 5 (16.7%) | |
| Missing | 15 (31.2%) | 0 (0%) | 0 (0%) | |
| **Diabetes** | | | | |
| No | 26 (54.2%) | 45 (95.7%) | 26 (86.7%) | 0.17 |
| Yes | 7 (14.6%) | 2 (4.3%) | 4 (13.3%) | |
| Missing | 15 (31.2%) | 0 (0%) | 0 (0%) | |
| **Duration of symptoms before admission (days)** | | | | |
| Mean (SD) | 4.06 (3.03) | 5.83 (5.65) | 4.70 (3.21) | 0.33 |
| Median [Min, Max] | 3.50 [1.00, 14.0] | 5.00 [1.00, 42.0] | 4.00 [0.00, 14.0] | |
| Missing | 16 (33.3%) | 0 (0%) | 0 (0%) | |
| **Antibiotic treatment before admission** | | | | |
| No | 27 (56.2%) | 29 (61.7%) | 23 (76.7%) | 0.17 |
| Yes | 5 (10.4%) | 18 (38.3%) | 7 (23.3%) | |
| Missing | 16 (33.3%) | 0 (0%) | 0 (0%) | |
| **Corticosteroid use before admission** | | | | |
| No | 29 (60.4%) | 46 (97.9%) | 29 (96.7%) | 0.67 |
| Yes | 2 (4.2%) | 1 (2.1%) | 1 (3.3%) | |
| Missing | 17 (35.4%) | 0 (0%) | 0 (0%) | |

Data are presented as number (%) or mean (SD). *Abbreviations*: PSI: pneumonia severity index; CNS: central nervous system; COPD: chronic obstructive pulmonary disease.

before admission. The analyses performed in this study were executed conform the informed consent given by the patients. The clinical data was anonymized before use.

## Bioanalytical procedures

Serum samples were analyzed with five liquid chromatography methods and one gas chromatography, mass spectrometry-based, targeted, metabolomics method. The metabolomics profiling covered 596 metabolite targets from 25 metabolite classes, including amino acids, biogenic amines, acylcarnitines, organic acids, and multiple classes of lipids (S3 Table). Levels of 374 unique metabolites were detected in the samples. The metabolomic profiling was performed within the Biomedical Metabolomics Facility of Leiden University in Leiden, The Netherlands. Details of the metabolomic analysis methods used are provided in S1 Method.

## Data analysis

The data resulting from the metabolomic profiling was cleaned by removing patient samples with more than 10 missing metabolite values, for example, if results from one measurement platform were missing because of too low sample volumes, and by removing metabolites with missing patient samples, for example, because of a sample preparation error. The clean dataset consisted of 347 metabolite levels (S4 Table) for 125 patients diagnosed with the microbial etiology *S. pneumoniae* (n = 48), atypical (n = 47), or viral (n = 30). The pathogens identified in each group are shown in Table 2. The resulting metabolite levels were preprocessed by applying log transformation and standardized to correct for heteroscedasticity. The preprocessed metabolomics dataset was visually inspected using a principal component analysis.

Data imputation was performed for patient characteristics that were to be evaluated as covariates in the statistical analysis and showed missingness in the data. Five times repeated imputation using predictive mean matching was performed with the 'mice' package for R to impute the patient data for the covariates with less than 25% missing data. Predictive mean matching is suitable for both numeric and binary covariates. Patient characteristics with >25% missing data were excluded from further analysis.

We performed logistic regression and elastic net regression modeling to determine if patients in one pathogen group could be discriminated from patients in the remaining two groups. Also, we aimed to determine which metabolites were important for prediction of the causative pathogen. In both methods, five-fold cross-validation was used to make the most efficient use of the available data for estimation of the predictive performance of the models and

**Table 2. Distribution of causative microbial agents per pathogen group for statistical data analysis.**

| Causative pathogen | S. pneumonia (n = 48) | Atypical bacterial (n = 47) | Viral (n = 30) |
|---|---|---|---|
| *S. pneumonia* | 48 (100%) | 0 (0%) | 0 (0%) |
| *Legionella pneumophila* | 0 (0%) | 18 (38.3%) | 0 (0%) |
| *Coxiella burnetii* | 0 (0%) | 17 (36.2%) | 0 (0%) |
| *Chlamydophila psittaci* | 0 (0%) | 7 (14.9%) | 0 (0%) |
| *Mycoplasma pneumoniae* | 0 (0%) | 5 (10.6%) | 0 (0%) |
| Influenza virus | 0 (0%) | 0 (0%) | 11 (36.7%) |
| HSV | 0 (0%) | 0 (0%) | 6 (20.0%) |
| RSV | 0 (0%) | 0 (0%) | 4 (13.3%) |
| Parainfluenza virus | 0 (0%) | 0 (0%) | 3 (10.0%) |
| Other viruses | 0 (0%) | 0 (0%) | 6 (20.0%) |

*Data are presented as number (%). Abbreviations*: *S. pneumoniae*: *Streptococcus pneumoniae; HSV: herpes simplex virus; RSV: respiratory syncytial virus.*

its associated metabolites [19]. Furthermore, the model generation was repeated 100 times to obtain robust estimates of the predictive performance of the models.

To identify single discriminative metabolites, Student's T-tests with false discovery rate (FDR) multiple testing corrections were performed (p < 0.05). Then, significant metabolites and a combination of significant metabolites were modeled using logistic regression. Also, models containing covariates age and sex and all covariates were generated. The predictive logistic regression models were analyzed by comparison of their area under the curve (AUC), sensitivity, specificity, balanced error rate (BER), and receiver operating characteristic (ROC) curve.

Elastic net regression was performed to test if the predictive power of the metabolite data could be increased by including correlations between metabolites in addition to evaluating single metabolites. In elastic net regression, metabolites that have no explanatory power can be set to zero, as in a lasso regression, and metabolites that explain the same amount of variance can all be included with balanced coefficient sizes, as in a ridge regression [20].

To obtain robust estimates of the predictive performance of the elastic net model, hyperparameters were optimized in a five-fold nested-cross validation, where the hyperparameters were selected truly independent of the calculation of the predictive performance, as is schematically shown in Fig 2 [21]. In the inner cross-validation loop, the model optimization loop, optimal values for model hyperparameters α and λ were determined. In the outer cross-validation loop, the model performance loop, the optimal model for the training fold was built on the set hyperparameters α and λ (S1 Fig). Hyperparameter selection was performed using the balanced error rate (BER), which can be calculated from the true- and false positive (TP, FP), and true- and false-negative rates (TN, FN, Eq 1). The BER accounts for different group sizes per model and therefore gives an accurate picture of the performance of models in the model optimization and model performance loop.

$$BER = 0.5 * \left( \frac{FP}{TN + FP} + \frac{FN}{FN + TP} \right) \tag{1}$$

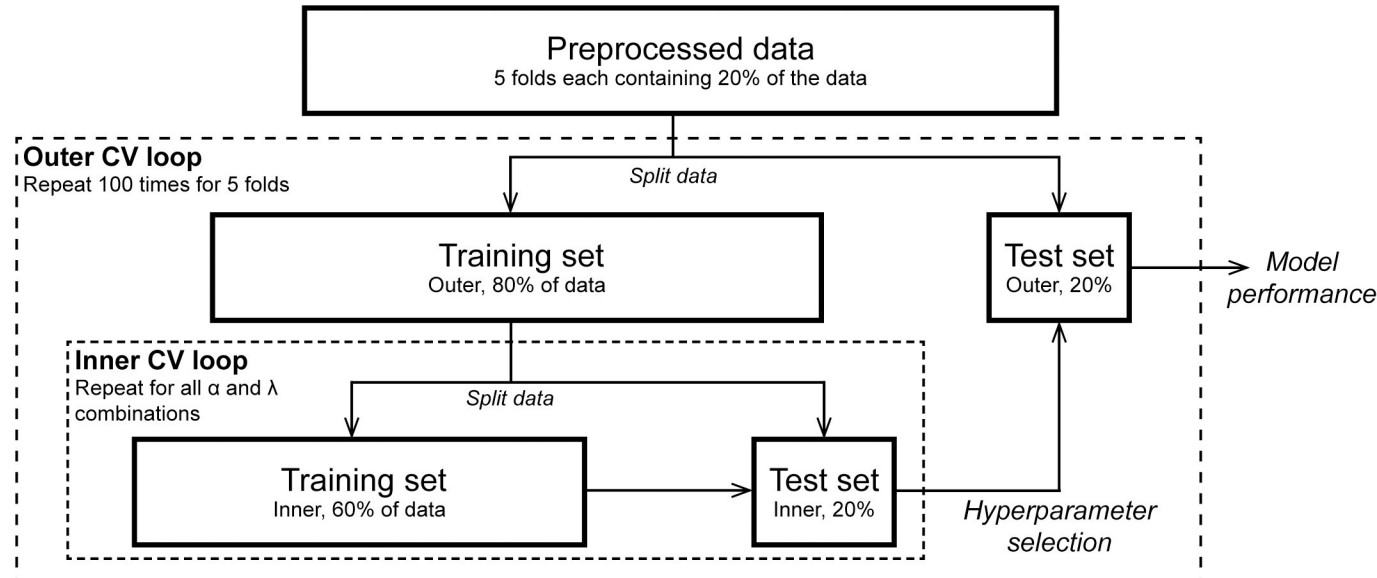

**Fig 2. Schematic representation of stratified nested cross-validation for elastic net regression model optimization and performance [21].** Abbreviations: CV: cross-validation.

The overall predictive diagnostic performance was evaluated using sensitivity and specificity performance measures, generated from the confusion matrix that represents the number of samples falling into each possible outcome (Eq 2–3). The average sensitivity and specificity of all 500 generated models and its standard deviation were used to compare the assay performance to currently used methods.

$$\text{Sensitivity} = \frac{\text{TP}}{\text{TP} + \text{FN}} \tag{2}$$

$$\text{Specificity} = \frac{\text{TN}}{\text{TN} + \text{FP}} \tag{3}$$

The relative contribution of metabolites to provide predictions of the expected pathogen group were quantified using the variable importance in prediction (VIP) score, expressed as a percentage. The VIP score was calculated per metabolite per fold or repeat as follows:

$$\text{VIP } (\%) = \frac{\beta_j}{\sum_{i=0}^{p} |\beta_i|} \cdot 100\% \tag{4}$$

where $\beta_j$ is the regression coefficient for fold j over the sum of all regression coefficient values in the model. Metabolites were arranged based on their mean VIP score over all folds and repeats. Metabolites with an absolute VIP > 1% were considered to be most important. Furthermore, to determine the need to include age and sex, or all covariates in the models we compared the BER for models with and without age and sex, or all covariates included. Finally, mean AUC values and ROC curves were calculated and generated to compare the performance of the elastic net models to the logistic regression models.

The scripts used for the statistical analyses were deposited in Github at http://github.com/vanhasseltlab/MetabolomicsEtiologyCAP.

## Results

### Metabolomics profiling and exploratory analysis of metabolomics data

Metabolomics profiling was performed for 130 patients and 596 metabolite targets. Preprocessing of the metabolomics dataset resulted in a reduced dataset including 125 patients and 347 metabolites (Fig 1). The patient characteristics of these 125 patients are displayed in Table 1. The patients were diagnosed with the microbial etiology *S. pneumoniae* (n = 48), atypical bacteria (n = 47), or respiratory virus (n = 30) (Table 2). A list of all targeted and detected metabolites and their identifiers can be found in S4 Table. Unsupervised principal component analysis showed no clear separation between pathogen groups (S2 Fig).

### Single discriminating metabolites for pathogen groups

Three significant metabolites were found for the discrimination of atypical pathogens from *S. pneumoniae* and viral pathogens using a Student's T-test with FDR multiple testing correction (p < 0.05): glycylglycine, symmetric dimethylarginine (SDMA), and lysophosphatidylinositol (18:1) (LPI (18:1)). For the other comparisons, no significantly discriminating metabolites were found.

The significantly differentiating metabolites were included in logistic regression models to differentiate patients with atypical pathogens from patients suffering from CAP caused by *S. pneumoniae* or viral pathogens. The logistic regression models were evaluated based on their AUC, sensitivity, specificity, BER, and ROC curve after fivefold cross-validation with 100

**Table 3. Results from the logistic regression and elastic net regression models that were tested in a fivefold cross-validation with 100 repeats.**

| Model | Variables | AUC | Sensitivity | Specificity | BER |
|---|---|---|---|---|---|
| *Atypical–(S. pneumoniae + viral)* | | | | | |
| Logistic Regression | Glycylglycine | 0.72 (0.094) | 0.36 (0.14) | 0.83 (0.110) | 0.40 (0.084) |
| Logistic Regression | SDMA | 0.72 (0.093) | 0.36 (0.15) | 0.86 (0.100) | 0.39 (0.082) |
| Logistic Regression | LPI.18.1. | 0.70 (0.099) | 0.32 (0.14) | 0.85 (0.100) | 0.41 (0.082) |
| Logistic Regression | Age + sex | 0.71 (0.097) | 0.39 (0.15) | 0.85 (0.090) | 0.38 (0.071) |
| Logistic Regression | All covariates | 0.65 (0.098) | 0.52 (0.15) | 0.68 (0.120) | 0.40 (0.087) |
| Logistic Regression | Glycylglycine + SDMA + LPI.18.1. | 0.78 (0.094) | 0.57 (0.16) | 0.83 (0.100) | 0.30 (0.090) |
| Logistic Regression | Glycylglycine + SDMA + LPI.18.1. + age + sex | 0.79 (0.089) | **0.63 (0.16)** | 0.84 (0.095) | **0.26 (0.085)** |
| Logistic Regression | Glycylglycine + SDMA + LPI.18.1. + all covariates | 0.75 (0.097) | 0.60 (0.16) | 0.78 (0.110) | 0.31 (0.093) |
| Elastic net regression | 100 (82) | **0.81 (0.087)** | 0.61 (0.18) | **0.86 (0.092)** | 0.27 (0.094) |
| Elastic net regression | 110 (91) incl. age & sex | 0.80 (0.094) | 0.61 (0.17) | 0.84 (0.096) | 0.28 (0.090) |
| Elastic net regression | 270 (140) incl. all covariates | 0.69 (0.100) | 0.58 (0.17) | 0.70 (0.120) | 0.36 (0.098) |
| *S. pneumoniae–(atypical + viral)* | | | | | |
| Elastic net regression | 210 (120) | **0.74 (0.091)** | **0.83 (0.10)** | 0.50 (0.160) | **0.33 (0.087)** |
| Elastic net regression | 240 (130) incl. age & sex | 0.74 (0.095) | 0.80 (0.10) | **0.52 (0.160)** | 0.34 (0.084) |
| Elastic net regression | 290 (120) incl. all covariates | 0.63 (0.110) | 0.69 (0.13) | 0.51 (0.17) | 0.40 (0.098) |
| *Viral–(S. pneumoniae + atypical)* | | | | | |
| Elastic net regression | 170 (140) | 0.54 (0.120) | 0.88 (0.11) | 0.16 (0.170) | 0.48 (0.075) |
| Elastic net regression | 130 (130) incl. age & sex | **0.63 (0.130)** | **0.89 (0.08)** | 0.23 (0.160) | **0.44 (0.082)** |
| Elastic net regression | 180 (160) incl. all covariates | 0.56 (0.130) | 0.79 (0.11) | **0.31 (0.190)** | 0.45 (0.099) |

The table displays the performance of the models for the three comparisons: atypical versus *S. pneumoniae* and viral pathogens; *S. pneumoniae* pathogens versus atypical and viral pathogens; and viral versus *S. pneumoniae* and atypical pathogens. Logistic regression is only included for the comparison of atypical versus S. pneumoniae and viral pathogens because no significant single metabolites were found for the other comparisons. The performance is evaluated using the mean area under the curve (AUC), the mean sensitivity, the mean specificity, and the mean balanced error rate (BER) over all folds and repeats. All performances result from the test sets within the cross-validation. The best performing model per comparison and evaluation measure is displayed in bold and underlined.

*Data are presented as mean (SD). Variables are presented as variable names or as the number of variables that are included in the model. Abbreviations: SDMA: symmetric dimethylarginine, LPI (18:1): lysophosphatidylinositol (18:1), AUC: area under the curve, BER: balanced error rate.*

repeats (Table 3, Fig 3). They show that logistic regression models of the individual metabolites glycylglycine, SDMA, and LPI(18:1) can differentiate atypical pathogens from *S. pneumoniae* and viral pathogens with AUCs between 0.70–0.72, sensitivities between 0.32–0.36, sensitivities between 0.83–0.85, and BERs of 0.39–0.41. A logistic regression model including all three significantly discriminating metabolites yields a more successful separation with an AUC of 0.78, sensitivity of 0.57, specificity of 0.83, and BER of 0.30. Addition of the covariates age and sex to the three metabolite model, slightly improved the predictive performance of the model resulting in a sensitivity of 0.63 and a specificity of 0.84. This model also showed the highest AUC (0.79) and lowest BER (0.26) of the tested logistic regression models. The addition of other covariates to the logistic regression model resulted in lower performance, probably due to overfitting of the model. The ROC curves emphasize the increased model performance upon the addition of more discriminating metabolites to the logistic regression model (Fig 3).

## Predictive metabolites for diagnosis of CAP-associated pathogens

Elastic net models including multiple metabolites were fit to discriminate *S. pneumoniae*, atypical bacterial, and viral pathogens from the remaining two groups (e.g., *S. pneumoniae* versus atypical bacterial and viral pathogens). Elastic net models separating patients with atypical

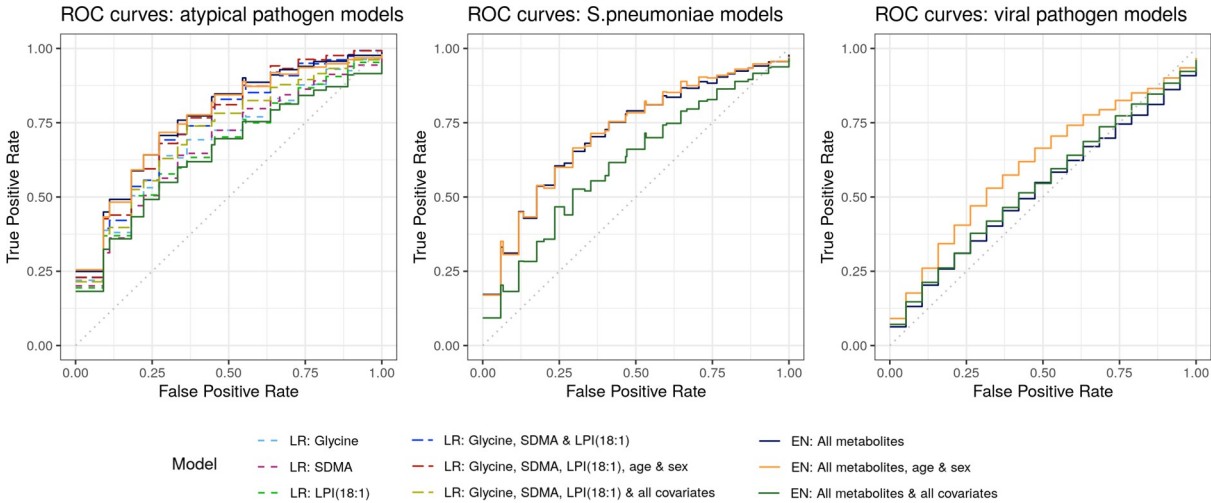

**Fig 3. ROC curves of the results from logistic regression and elastic net regression models that were tested in five-fold cross-validation with 100 repeats for the comparisons: atypical versus S. pneumoniae and viral pathogens; S. pneumoniae pathogens versus atypical and viral pathogens; and viral versus S. pneumoniae and atypical pathogens.** Abbreviations: LR: logistic regression, EN: elastic net regression, SDMA: symmetric dimethylarginine, LPI (18:1): lysophosphatidylinositol (18:1).

bacterial pathogens from patients with *S. pneumoniae* and viral infections resulted in a mean AUC of 0.81, a sensitivity of 0.61, a specificity of 0.86, and a BER of 0.26. Prediction of *S. pneumoniae* or viral infection etiologies showed lower predictive capabilities with AUC's of 0.74 and 0.63, high sensitivities of 0.83 and 0.89, but low specificities of 0.5 and 0.23, and BER's of 0.33 and 0.44, respectively (Table 3).

We included the covariates age and sex, and all covariates in the elastic net models to account for potential confounding effects. The addition of these covariates showed no improved performance of the elastic net models for differentiation of atypical pathogens or *S. pneumoniae* from the other groups. For the differentiation of viral pathogens from the other two pathogen groups, a slight performance improvement was seen upon the addition of the covariates age and sex resulting in an AUC of 0.63, a sensitivity of 0.89, a specificity of 0.23, and a BER of 0.44 (Table 3).

The ROC curves for the separation of atypical pathogens from *S. pneumoniae* and viral pathogens show that elastic net models perform better than the logistic regression models for single metabolites. However, the logistic regression model including the three significant metabolites and the covariates age and sex shows similar performance as the elastic net regression which included 100 metabolites on average (Fig 3).

## Metabolite classes predictive for atypical bacterial pathogens

Focusing on the metabolites that have shown to be predictive for atypical bacterial pathogens, i.e., the only comparison with clinically relevant predictive performance, we identified 26 metabolites with an absolute VIP > 1% using elastic net regression (Fig 4). The metabolites originated from multiple metabolite classes. However, the classes of biogenic amines and lysophospholipids were well represented (4–5 metabolites per class), compared to the other classes. The number of metabolites included in the models varied across folds without a clear correlation to the BER. Commonly, models including all metabolites were favored, followed by models including 20–100 metabolites (S3 Fig). We visualized the separation of the different pathogens in the atypical pathogen group using an unsupervised PCA analysis including all

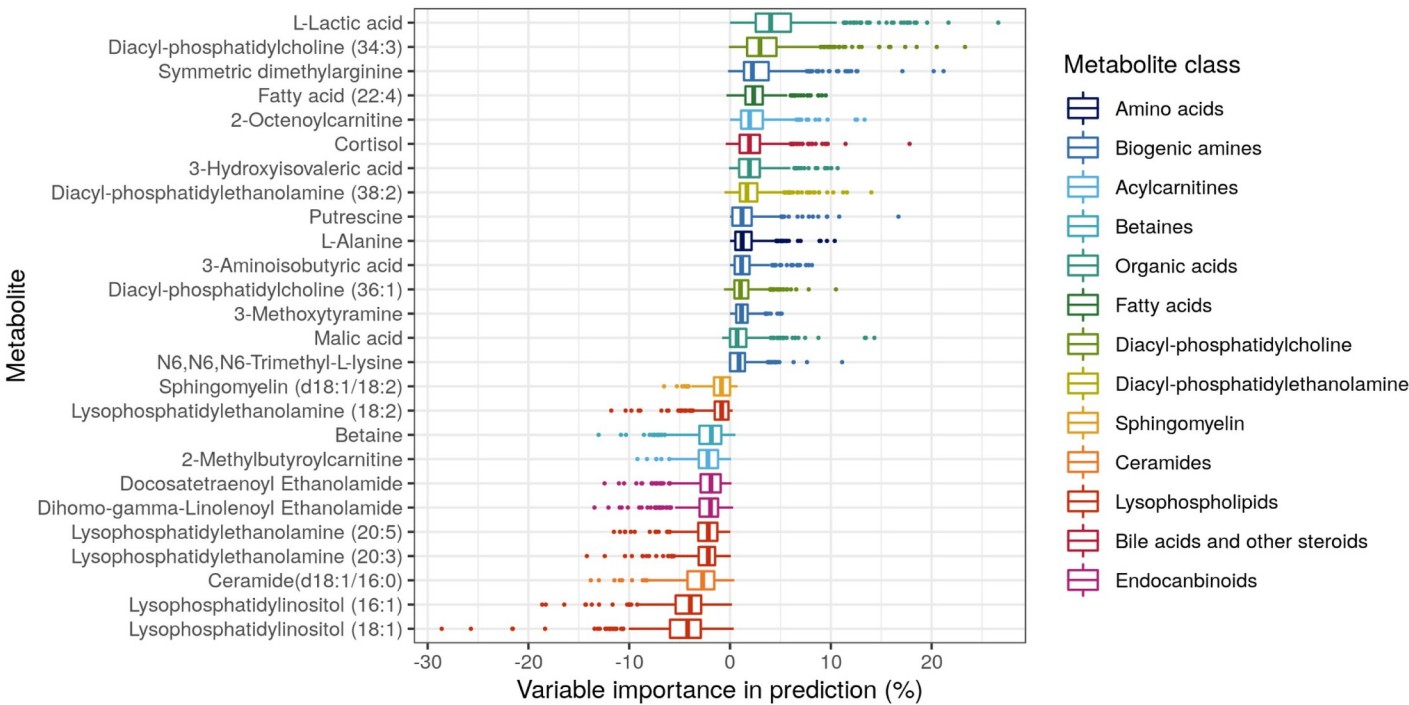

**Fig 4. Variable importance of metabolites for the prediction of an atypical bacterial infection versus *S. pneumoniae* and viral infections.** Only metabolites with an absolute mean percentage of influence > 1% are visualized.

metabolites. The PCA plot indicated that no clear sub-group is present within the atypical group that would prominently drive the separation from the *S. pneumoniae* and viral infections (S4 Fig).

## Discussion

Targeted profiling of the host metabolic response revealed metabolites that can support the diagnosis of microbial etiology in CAP patients with atypical bacterial pathogens compared to patients with *S. pneumoniae* or viral infections. CAP patients suffering from *S. pneumoniae* and viral infection could not be as successfully discriminated from the other groups based on the metabolic host-response.

The currently used clinical assays still outperform the metabolomics host-response assays developed in this study. For atypical pathogens, the sensitivity of 63% and specificity of 86% reported in this study are lower than the current urinary antigen tests for detection of *Legionella pneumophila* which shows a sensitivity of approximately 70% and a specificity up to 96% [22]. For detection of *S. pneumoniae*, the 83% sensitivity reached with the metabolomics-based assay outperforms the current antigen tests that show 70% sensitivity. However, the specificity of the metabolomics-based assay is only 50% while antigen tests reach specificity up to 96% [23, 24]. PCR assays of nasopharyngeal swabs for viral pathogens show sensitivities of up to 96% for influenza viruses A and B [25]. Our viral metabolomics-based assay shows a good sensitivity of 89% as well. However, the specificity of this assay is with 23% very low. The expected clinical utility of the studied metabolite classes as host-response biomarkers for etiological diagnosis of CAP may therefore be considered limited.

The combination of the metabolites glycylglycine, SDMA, and LPI (18:1) and the covariates age and sex showed predictive capacities similar to elastic net models including 100

metabolites in the comparison of atypical pathogens versus *S. pneumoniae* and viral pathogens. This result suggests that a simple model might perform as well as a more complex elastic net model, which is an important finding when considering the use of these biomarkers for clinical diagnostic applications, e.g., where a limited set of 3 metabolites is preferable.

Glycylglycine, a biogenic amine, showed to be significantly contributing to the differentiation of atypical pathogens from the other pathogens, but was not often included in elastic net models. In contrast, SDMA and LPI (18:1) were often included in the elastic net models as was shown in the overview of the 26 most influential metabolites. Metabolites of the classes biogenic amines and lysophospholipids, to which SDMA and LPI (18:1) have been assigned, were most represented in the 26 most influential metabolites compared to other metabolite classes in the comparison of atypical versus *S. pneumoniae* and viral pathogens. A comparison of the most influential metabolites in this study to metabolites of interest reported in previous studies of metabolomics in CAP patients shows limited overlap. Major reasons for this could be that (i) not all studies measured the same set of metabolic classes; (ii) some other studies poorly controlled patient comparator groups; and (iii) difference in bioanalytical methodologies, e.g. the use of NMR or MS as analytical method with their respective (dis)advantages might provide different results [26]. For example, most lipids found to be predictive in this study have not been reported previously, most likely because the applied bioanalytical methodologies did not allow their detection. However, some overlap was found between the most influential metabolites for the comparison of atypical versus *S. pneumoniae* and viral pathogens in this study, and the metabolites of interest from other metabolomics studies involving CAP patients. The amino acid alanine was found in multiple studies [14, 16, 17]. Ceramide (d18:1/16:0), two diacyl-phosphatidylcholines, and diacyl-phosphatidylethanolamine (38:2) were found in other studies as well, the latter in the form of choline and ethanolamine [15, 16, 18]. Lactic acid was identified by several other metabolomics studies to respiratory bacterial and viral infections [12, 14, 17]. Lactic acid levels are also known to rise in case of severe disease. However, because the three pathogen groups were balanced in terms of disease severity and, for example, did not show significant differences in pH levels, we hypothesize that the differences in lactate levels are, in this case, an effect of the pathogen-specific host-response to infection. The result showed that models including disease severity covariates do not perform better than models without these confounders, thus supporting this hypothesis. Finally, 3-hydroxyisovaleric acid and betaine have been reported in a previous study comparing viral and bacterial pneumonia [18]. The overlap in these findings may provide insights into common metabolic responses to pathogens involved in CAP.

Multiple biological processes besides infection can influence metabolic processes in patients. Inclusion of age and sex in the models did not improve the predictive performance of the elastic net models for atypical bacteria and *S. pneumoniae* but did improve the model for viral pathogens. The average age in the viral pathogen group was higher than in the other groups, which could explain this result. For the other comparisons, we see that a model including age and sex or more covariates does not outperform models without these possible confounders. This doesn't imply there is no metabolomic effect of age in the bacterial pathogen groups but implies that the separation between bacterial pathogen groups is more dependent on the metabolomic host-response to the infection than on the age-related metabolomic changes. In this study, we included patients with mild to severe CAP, reflecting the target patient population for which improvements in a diagnostic assay are required. However, the combination of samples from patients with different disease severities may negatively influence the predictive capabilities of the model because the effect from the causative pathogen on the host-metabolism may be less pronounced for less severe disease [27]. However, separating the patients into groups with comparable disease severity scores would decrease the power for

statistical analysis. Furthermore, no standardization of sampling times and conditions was applied, e.g., patients had not fasted before blood sampling, which may influence the metabolite patterns found. Since variations in sampling conditions were unknown, we were unable to consider these in our analyses. However, we expect that the impact of not standardizing and correcting for these factors is limited because the noise in metabolite levels introduced by these factors is expected to be random with regard to the pathogen groups compared in this study. A standardized sampling approach could improve the sensitivity of the models to detect predictive metabolites because some noise is reduced. However, the specificity of the models with respect to the prediction of specific pathogens would be unchanged, since no correlation with pathogen groups is likely.

The sample size of this study (n = 125) was relatively large compared to studies researching metabolomic differences between causative pathogens of CAP that included approximately 70 patients [17, 18]. The compared groups S. pneumoniae, atypical bacteria, and viruses were chosen because antibiotic treatment strategies differ between these three groups. Ideally, we would have further investigated differences within studied groups, e.g. to identify metabolic responses to specific pathogens within the atypical pathogens and viral infection groups. For example, it would be of interest to study Legionella species more in-depth because their intracellular growth might result in a differentiated host-response. However, this was considered not feasible in this study due to sample size restrictions. The heterogeneous pathogen population in the atypical bacterial and viral pathogen groups might have lowered the predictive performance of the metabolomic analysis. Studying the individual pathogens in bigger sample sizes might reveal more characteristic metabolite signatures. In this study, no control group was included because the goal of the study was to provide a faster and optimal diagnostic method and a guide for antibiotic treatment in hospitalized CAP patients. In further studies, it would be preferable to include patients with all causes of CAP, including the remaining microorganisms, which were excluded in the current study because of their low frequency, to enable a more comprehensive comparison with current clinical assays. In this study, CAP patients with unknown pathogens were excluded. In a follow-up study, the metabolite pattern of the patients with unknown causative pathogens could be compared to the metabolite patterns of the distinguished pathogen groups to gain more information about the metabolomic resemblance of the samples in which pathogens could and could not be identified using the conventional diagnostic techniques.

Metabolomics analysis resulted in some missing data because of sample preparation errors or the limited volume of the samples. Because the measurement platforms covered multiple metabolites within one pathway, metabolites with missing data could be removed without influencing the final results. Some patient samples had to be removed because of multiple missing metabolite levels, for example, if the results from a whole metabolomics platform were missing. Data imputation was not performed for the metabolomics data, because the wide range of patients included in the dataset did, in our opinion, not provide enough information for accurate data imputation.

In summary, this comprehensive analysis of the host metabolic response across multiple metabolic classes and based on a well-balanced study cohort of CAP patients has shown the possibility to identify atypical pathogens in CAP and limited utility of predicting *S. pneumoniae* and viral infection disease etiologies.

## Supporting information

**S1 Method. Details on metabolomic sample analysis.**
(DOCX)

**S1 Fig. Optimization of α and λ in the inner Cross-Validation (CV) to reach a minimal Balanced Error Rate (BER) in the outer CV.** (A) Shows all α and λ values tested in inner CV against mean BER of the inner CV. (B) A plot of the optimal α and λ combinations chosen in the inner CV against their BER in the outer CV shows a variety of favorable α and λ concentrations. (C) A plot of the number of variables selected in the elastic net model in outer CV shows that with increasing alpha, the number of variables decreases as is expected in an elastic net model. The data shown in the Fig is a result of the comparison Atypical–(S. pneumoniae + viral).
(DOCX)

**S2 Fig. Unsupervised Principal Component Analysis (PCA) plot of all pathogen groups.**
(DOCX)

**S3 Fig.** (A) Boxplot of BER per number of variables selected shows no clear relation between the number of variables selected and model performance. (B) Histogram of the number of variables selected shows that a model with all metabolites included is favored, followed by models including 34, 49, 82, 24, or 45 metabolites. Both Figs contain the data of all folds and repeats (n = 500) for the comparison between atypical versus S. pneumoniae and viral infections.
(DOCX)

**S4 Fig. Principal Component Analysis (PCA) of the atypical pathogen group (log-transformed and standardized data) shows that there is no clear subgroup within the atypical group that would prominently drive the separation from the S. pneumoniae and viral infections.**
(DOCX)

**S1 Table. Summary of previous studies focusing on bacterial and viral respiratory tract infections and related metabolites.**
(DOCX)

**S2 Table. Additional patient characteristics per pathogen group.**
(DOCX)

**S3 Table. Overview of the number of metabolites included in the metabolomics platforms, measured in the samples and included in the data analysis.**
(DOCX)

**S4 Table. Information on measurement platforms used, metabolite classes targeted per platform, targeted metabolites, their abbreviations and names in R (if detected) and identifiers (if available).**
(XLSX)

**S5 Table. Metabolomics data after quality control.**
(CSV)

## Author Contributions

**Conceptualization:** Thomas Hankemeier, Ewoudt M. W. van de Garde, J. G. Coen van Hasselt.

**Data curation:** Ilona den Hartog, Stefan M. T. Vestjens, Amy C. Harms.

**Formal analysis:** Ilona den Hartog, Laura B. Zwep.

**Funding acquisition:** Amy C. Harms, G. Paul Voorn, Dylan W. de Lange, Willem J. W. Bos, Thomas Hankemeier, Ewoudt M. W. van de Garde, J. G. Coen van Hasselt.

**Investigation:** Ilona den Hartog.

**Methodology:** Ilona den Hartog, J. G. Coen van Hasselt.

**Project administration:** Ilona den Hartog, Amy C. Harms, J. G. Coen van Hasselt.

**Resources:** Stefan M. T. Vestjens, Amy C. Harms, Willem J. W. Bos, Thomas Hankemeier, Ewoudt M. W. van de Garde, J. G. Coen van Hasselt.

**Software:** Ilona den Hartog, Laura B. Zwep.

**Supervision:** Thomas Hankemeier, Ewoudt M. W. van de Garde, J. G. Coen van Hasselt.

**Validation:** Ilona den Hartog, Laura B. Zwep, Amy C. Harms.

**Visualization:** Ilona den Hartog.

**Writing – original draft:** Ilona den Hartog.

**Writing – review & editing:** Laura B. Zwep, Stefan M. T. Vestjens, Amy C. Harms, G. Paul Voorn, Dylan W. de Lange, Willem J. W. Bos, Thomas Hankemeier, Ewoudt M. W. van de Garde, J. G. Coen van Hasselt.

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
