## [Decision Letter · Decision Letter 0]

1 Feb 2021

PONE-D-21-00435

Metabolomic profiling of microbial disease etiology in community-acquired pneumonia

PLOS ONE

Dear Dr. den Hartog,

Thank you for submitting your manuscript to PLOS ONE. After careful consideration, we feel that it has merit but does not fully meet PLOS ONE’s publication criteria as it currently stands. Therefore, we invite you to submit a revised version of the manuscript that addresses the points raised during the review process.

We look forward to receiving your revised manuscript.

Kind regards,

Aran Singanayagam

Academic Editor

PLOS ONE

Journal Requirements:

Reviewers' comments:

Reviewer's Responses to Questions

**Comments to the Author**

1. Is the manuscript technically sound, and do the data support the conclusions?

Reviewer #1: Yes

Reviewer #2: Yes

2. Has the statistical analysis been performed appropriately and rigorously? 

Reviewer #1: I Don't Know

Reviewer #2: Yes

3. Have the authors made all data underlying the findings in their manuscript fully available?

Reviewer #1: Yes

Reviewer #2: Yes

4. Is the manuscript presented in an intelligible fashion and written in standard English?

Reviewer #1: Yes

Reviewer #2: Yes

5. Review Comments to the Author

Reviewer #1: The authors undertook a metabolomics study of serum samples in hospitalized CAP patients to determine if host-response associated metabolites can enable diagnosis of microbial etiology. The topic is relevant.

The authors conclude that' targeted profiling of the host metabolic response revealed metabolites that can support diagnosis of microbial etiology in CAP patients with atypical bacterial pathogens compared to patients with S. pneumoniae or viral infections'. However, they also admit that 'the currently used clinical assays still outperform the metabolomics host-response assays developed in this study'. Despite the later, the study is still sound I guess. It is challenging to get homogenous groups and, depending on the sampling and method used, different results are expected. This makes a generalizability challeninge for this study.

I probably missed this point. When were the serum samples taken? In the morning before food? Has this been standardized?

Table 1: A pity that BMI has not been known. What about use of antibiotics? Ethnicity? Authors could add p-values so one would see if some of the patient characteristics per pathogen group are different between the three groups. E.g. is age really significnatly higher in the s. pneumoniae group?

Table 2: The 'atypical bacterial' and 'viral' groups are still heterogenous. Has this been considered in the models? Different atypical bacteria may result in a different profile? Is there a reason why the authors didn't include some individuals without CAP (control group)?

The authors did not use NMR. Could the authors elaborate on the pros and cons of using NMR as compared to their methods? They only mention 'reduced sensitivity', but there are also advantages using nmr. It seems to me that different methods lead to different conclusions.

No line numbers in the discussion.

Reviewer #2: I am reviewing the article titled “Metabolomic profiling of microbial disease etiology in community-acquired pneumonia” by den Hartog et al. These investigators performed a “large-scale” metabolomics study from human serum samples of severe pneumonia (necessitating hospital admission). The researchers focused on three distinct groups of patients those with S. pneumoniae, atypical bacteria, or viral infections. The authors utilized multiple methods to determine discriminating metabolites, they found that there is a possible method to determine atypical infections from S. pneumoniae vs. viral pathogens.

Strengths

The authors have extensively profiled sick patients with pneumonia through an untargeted metabolomic profile. The authors use extensive statistical modeling using these tools to determine if there are differences between the patients with three different pneumonias. They found that there are ways to determine the differences between atypical pneumonias compared to Streptococcus/viral pneumonias. However, these methods are not sensitive nor specific enough compared to approved clinical tests. The work is thorough and well-documented; however I believe that it is missing some clinical relevance.

Weaknesses

As a proof of concept, this is an excellent manuscript, but I am less enthusiastic for several reasons: 1) As a clinician, several significant clinical outcomes of interest, including things like antibiotics, oxygen requirement, and if the patients were sick or not sick. If we are talking about host-response, these factors may play a bigger role and may confound their analysis, 2) Lumping severity into a score (e.g., PSI), 3) Other medications and intrinsic lung disease are not mentioned as possible contributors to their model, 4) clinical relevance, if clinicians and researchers are able to tell the difference between certain infections, then what can utilizing a metabolomic approach offer a researcher or clinician? Finally, 5) Was there another testing cohort to test their model?

Two interesting points that may be beyond the scope of the work by the authors: 1) Was there ever thought about comparing the metabolites to healthy subjects compared to pneumonia subjects? 2) Although there is little difference between the atypical pneumonia pathogens, there almost appears to be a distinct group between the legionella compared to mycoplasma samples. Was there thought about exploring possible differences between these two groups?

I will be using the page number and the left most line numbers. In the discussion section there doesn’t seem to be line numbers.

Major

Introduction:

Page 10, Line 68 “The studies that compared viral and bacterial …” I would just be careful and call this a limitation. Untargeted metabolomics may offer significant benefits in terms of identifying unknown metabolites. An untargeted approach is much more similar to a fishing expedition, I agree, but there may be some benefits compared to a targeted approach.

Materials and methods:

Page 11, Line 95 “The study …” One question I was wondering that the authors may have addressed at a different point was the length of time related to the patient’s illness? While it’s interesting that these patients all felt ill enough to come into the hospital, it’s not quite clear if the length of time they were sick would have confounded their analysis. For example, a person sick enough to come to the hospital on day 5 may be different than one that arrives 14 days after falling ill.

Page 11, Line 99 There is very limited clinical information that would confound host-metabolite expression, for example 1) Use of supplemental oxygen? 2) Other comorbid disease states such as diabetes, 3) BMI (which the authors mentioned in the conclusion was not recorded), 4) medications the patient had been taken prior to “catching” pneumonia (e.g., steroids, inhalers, antibiotics), and 5) most interesting of all, no mention of pre-existing lung disease (e.g., COPD, asthma, ILD). For host-metabolite issues, these would be of interesting to understand if they impact host-expression, especially lung and systemic metabolites.

Page 12, Line 133 “… models containing age and sex were generated …” Given the predilection of Streptococcus pneumonia impact older subjects, I am a little surprised that age did not factor into the analysis as in Table 1 it seems as though the age would be statistically different.

Results

Page 15, Line 189 “Single discriminating metabolites for pathogen groups”. Out of curiosity and this may be beyond the scope of the study, was there any distinct groups that were identified in an unsupervised fashion? From the metabolites, could the authors identify distinct groups? I am wondering if using the data to find distinct groups could also be performed (again beyond the scope of the study, but could be interesting to look at to see if there may be groups that are not clearly seen). For example, using Dirichlet Multinomial Mixtures to identify distinct groups. This could be added as a figure in the supplement. Part of me wonders if differences in serum metabolites may be associated with clinical outcomes.

Discussion

Page 20, Line … “Targeted …” I appreciate that the authors point out that it is difficult based on the host-metabolomic profile to tell the difference between the various pneumonias. What isn’t clear to me is why would atypical infections, in particular have such distinct host-metabolomic profile? The authors do a commendable effort into searching for metabolites which can discriminate between infections, but what is so particular that the infections create a unique host response (e.g., such as the intra-cellular nature of some of these infections Mycoplasma and Legionella).

Page 21, Line … “Lactic acid …” I think this is interesting because there are R and L enantiomers that are involved in microbial metabolism, but from a clinical point of view, lactemia in the serum is sign of severe disease. Perhaps, it may actually reflect severity of disease.

Page 22, Line … “In this study, we included patients …” It’s interesting that the authors utilized a pneumonia score, perhaps to understand some of the granularity of the data the authors should try to expand the PSI score and reassess their model based on the severity of disease? Moreover, have the authors tried to separate out the analysis based upon severity? The severity of disease could serve as a confounder in their analysis. I recommend the authors split the PSI score and attempt to construct their models utilizing

Minor

None

6. PLOS authors have the option to publish the peer review history of their article (what does this mean?). If published, this will include your full peer review and any attached files.

Reviewer #1: No

Reviewer #2: No

---

## [Author Response · Author response to Decision Letter 0]

10 Mar 2021

Dear dr. Singanayagam and reviewers, 

We would like to thank you for thoroughly reading our manuscript and providing us with constructive feedback. We revised the manuscript based on your comments. We made adjustments to clarify certain sections and improve the quality of the manuscript. We hope to have addressed all concerns to your satisfaction. Below, we respond to each comment individually. The line numbers we added refer to the clean version of the revised manuscript. 

Reviewer #1:

1. I probably missed this point. When were the serum samples taken? In the morning before food? Has this been standardized?

Since the samples were not collected specifically for metabolomics analysis, no standardization for sampling times and conditions was applied. In the Methods section we stated the only available information (lines 85-86): 

“The samples were taken from CAP patients within 24 hours after hospital admission.” 

2. Table 1: A pity that BMI has not been known. What about use of antibiotics? Ethnicity? Authors could add p-values so one would see if some of the patient characteristics per pathogen group are different between the three groups. E.g. is age really significnatly higher in the s. pneumoniae group?

We agree with the reviewer that it would have been beneficial to have known the BMI, since it is known to influence metabolomic profiles. We added extra information to Table 1 (line 113): P-values to visualize significant differences between groups, COPD, diabetes, duration of symptoms before admission, antibiotic treatment before admission, corticosteroid use before admission. We have added S2 Table (line 612) with additional patient characteristics as well, containing: race, nursing home resident, altered mental status, respiratory rate, systolic blood pressure, temperature, pulse, pH, BUN, sodium, glucose, hematocrit, partial pressure of oxygen, oxygen saturation, supplemental oxygen required.

The mean age is indeed significantly different between the three groups as we expected based on literature. This is, for example, described by Raeven et al., BMC Infect Dis. 2016 June 17; 16(299). 

3. Table 2: The 'atypical bacterial' and 'viral' groups are still heterogenous. Has this been considered in the models? Different atypical bacteria may result in a different profile? 

This heterogeneous character of the atypical bacterial and viral groups has not been taken into account in the models. We have added a clarification to the Discussion section:

Lines 354-355: “The compared groups S. pneumoniae, atypical bacteria, and viruses were chosen because antibiotic treatment strategies differ between these three groups.”

Lines 360-362:“The heterogeneous pathogen population in the atypical bacterial and viral pathogen groups might have lowered the predictive performance of the metabolomic analysis. Studying the individual pathogens in bigger sample sizes might reveal more characteristic metabolite signatures.” 

4. Is there a reason why the authors didn't include some individuals without CAP (control group)? 

We have emphasized our approach more clearly in the Discussion (lines 362-364):

“In this study, no control group was included because the goal of the study was to provide a faster and optimal diagnostic method and a guide for antibiotic treatment in hospitalized CAP patients.”

For more general, biological analysis, we think that the inclusion of a control group could be of interest to provide more insight into the metabolomic differences between healthy and diseased individuals. 

5. The authors did not use NMR. Could the authors elaborate on the pros and cons of using NMR as compared to their methods? They only mention 'reduced sensitivity', but there are also advantages using nmr. It seems to me that different methods lead to different conclusions.

We rephrased the reference to the difference between NMR and MS methods in the Discussion (lines 314-317):

“Major reasons for this could be that (i) not all studies measured the same set of metabolic classes; (ii) some other studies poorly controlled patient comparator groups; and (iii) difference in bioanalytical methodologies, e.g. the use of NMR or MS as analytical method with their respective (dis)advantages might provide different results [26]”

6. No line numbers in the discussion.

Something went wrong indeed. We have added line numbers in the discussion in the entire manuscript. 

Reviewer #2: 

1. As a clinician, several significant clinical outcomes of interest, including things like antibiotics, oxygen requirement, and if the patients were sick or not sick. If we are talking about host-response, these factors may play a bigger role and may confound their analysis. Lumping severity into a score (e.g., PSI), Other medications and intrinsic lung disease are not mentioned as possible contributors to their model, 

We would like to thank the reviewer for these suggestions. We have added the clinical parameters that were available for this study to the patient characteristics table (Table 1 & S2 Table). To Table 1 we have added: COPD, Diabetes, Duration of symptoms before admission, antibiotic treatment before admission, corticosteroid use before admission. In S2 Table we have included: race, nursing home resident, altered mental status, respiratory rate, systolic blood pressure, temperature, pulse, pH, BUN, sodium, glucose, hematocrit, partial pressure of oxygen, oxygen saturation, supplemental oxygen required. 

In the materials and methods section we have defined which confounders were included in the model (lines 102-106): 

“Patient characteristics that might be considered as possible covariates were: age, sex, nursing home resident, renal disease, congestive heart failure, CNS disease, malignancy, COPD, diabetes, altered mental status, respiratory rate, systolic blood pressure, temperature, pulse, pH, BUN, sodium, glucose, hematocrit, partial pressure of oxygen, pleural effusion on x-ray, duration of symptoms before admission, antibiotic treatment before admission.”

For the remaining patient characteristics, there was 100% the same value in all samples, for example there were no patients with liver disease, or >25% missingness in the data. We clarified the imputation procedure in the method (lines 135-140): 

“Data imputation was performed for patient characteristics that were to be evaluated as covariates in the statistical analysis and showed missingness in the data. Five times repeated imputation using predictive mean matching was performed with the ‘mice’ package for R to impute the patient data for the covariates with less than 25% missing data. Predictive mean matching is suitable for both numeric and binary covariates. Patient characteristics with >25% missing data were excluded from further analysis.”

The results of adding extra confounders to the logistic regression and elastic net models are presented in Fig. 3 and Table 3 and in writing in the Results section:

Lines 225-226: “The addition of other covariates to the logistic regression model resulted in lower performance, probably due to overfitting of the model.”

Lines 237-242: “We included the covariates age and sex, and all covariates in the elastic net models to account for potential confounding effects. The addition of these covariates showed no improved performance of the elastic net models for differentiation of atypical pathogens or S. pneumoniae from the other groups. For the differentiation of viral pathogens from the other two pathogen groups, a slight performance improvement was seen upon the addition of the covariates age and sex resulting in an AUC of 0.63, a sensitivity of 0.89, a specificity of 0.23, and a BER of 0.44 (Table 3).”

2. If clinicians and researchers are able to tell the difference between certain infections, then what can utilizing a metabolomic approach offer a researcher or clinician? 

This is however currently not the case. As we stated in the introduction (lines 51-52) 

“In over 60% of CAP patients, no causative pathogen can be identified with these pathogen-targeted diagnostic techniques [2,6]”. 

Identifying microbial diagnosis for this patient group could improve patient care by guiding antibiotic therapy and possibly reduce the risk for the development of antimicrobial resistance. 

3. Was there another testing cohort to test their model?

No, no separate testing cohort was available. We chose to use a nested cross-validation approach to validate our model as is explained in the method section (lines 159-164) and Fig. 2 (line 196). 

Two interesting points that may be beyond the scope of the work by the authors: 

4. Was there ever thought about comparing the metabolites to healthy subjects compared to pneumonia subjects? 

We have emphasized our approach more clearly in the Discussion (lines 362-364):

“In this study, no control group was included because the goal of the study was to provide a faster and optimal diagnostic method and a guide for antibiotic treatment in hospitalized CAP patients.”

For more general, biological analysis, we think that the inclusion of a control group could be of interest to provide more insight into the metabolomic differences between healthy and diseased individuals.

5. Although there is little difference between the atypical pneumonia pathogens, there almost appears to be a distinct group between the legionella compared to mycoplasma samples. Was there thought about exploring possible differences between these two groups?

We have not attempted to separate individual pathogens with predictive modeling and have added a clarification about this topic to the Discussion section: 

Lines 354-355: “The compared groups S. pneumoniae, atypical bacteria, and viruses were chosen because antibiotic treatment strategies differ between these three groups.”

Lines 360-362: “The heterogeneous pathogen population in the atypical bacterial and viral pathogen groups might have lowered the predictive performance of the metabolomic analysis. Studying the individual pathogens in bigger sample sizes might reveal more characteristic metabolite signatures.” 

Lines 355-360: “Ideally, we would have further investigated differences within studied groups, e.g. to identify metabolic responses to specific pathogens within the atypical pathogens and viral infection groups. For example, it would be of interest to study Legionella species more in-depth because their intracellular growth might result in a differentiated host-response. However, this was considered not feasible in this study due to sample size restrictions.”

6. Introduction: Page 10, Line 68 “The studies that compared viral and bacterial …” I would just be careful and call this a limitation. Untargeted metabolomics may offer significant benefits in terms of identifying unknown metabolites. An untargeted approach is much more similar to a fishing expedition, I agree, but there may be some benefits compared to a targeted approach.

We have rephrased this sentence in the introduction to underline the benefits of both untargeted and targeted metabolomics (lines 68-72): 

“The studies that compared viral and bacterial causative pathogen groups of CAP used an untargeted metabolomics approach. While an untargeted approach is especially useful for the discovery of new metabolites and hypothesis-free analysis, a targeted approach that can be fully quantified to clinical laboratory standards may be preferable for clinical implementation.”

7. Materials and methods: Page 11, Line 95 “The study …” One question I was wondering that the authors may have addressed at a different point was the length of time related to the patient’s illness? While it’s interesting that these patients all felt ill enough to come into the hospital, it’s not quite clear if the length of time they were sick would have confounded their analysis. For example, a person sick enough to come to the hospital on day 5 may be different than one that arrives 14 days after falling ill.

We agree and have added the variable “Duration of symptoms before admission” to the patient characteristics and our models (see the response in point 1, reviewer 2). We did not see a significant difference between the three groups, but we cannot exclude that some noise might have been introduced. 

8. Materials and methods: Page 11, Line 99 There is very limited clinical information that would confound host-metabolite expression, for example 1) Use of supplemental oxygen? 2) Other comorbid disease states such as diabetes, 3) BMI (which the authors mentioned in the conclusion was not recorded), 4) medications the patient had been taken prior to “catching” pneumonia (e.g., steroids, inhalers, antibiotics), and 5) most interesting of all, no mention of pre-existing lung disease (e.g., COPD, asthma, ILD). For host-metabolite issues, these would be of interesting to understand if they impact host-expression, especially lung and systemic metabolites.

We have added all available patient characteristics related to these possible confounders (see point 1 in response to reviewer 2). There are no significant differences for patient characteristics between the three compared pathogen groups, except for age. 

9. Materials and methods: Page 12, Line 133 “… models containing age and sex were generated …” Given the predilection of Streptococcus pneumonia impact older subjects, I am a little surprised that age did not factor into the analysis as in Table 1 it seems as though the age would be statistically different.

In the studied patient cohort, the mean age was significantly higher in patients with viral CAP compared to bacterial CAP. We hypothesized that this age difference could confound the results but our results show that this is not the case in our cohort. To clarify, we have added to the discussion section (lines 339-343): 

“We see that a model including age and sex does not outperform models without these possible confounders. This doesn’t imply there is no metabolomic effect of age in the bacterial pathogen groups but implies that the separation between bacterial pathogen groups is more dependent on the metabolomic host-response to the infection than on the age-related metabolomic changes.”

10. Results: Page 15, Line 189 “Single discriminating metabolites for pathogen groups”. Out of curiosity and this may be beyond the scope of the study, was there any distinct groups that were identified in an unsupervised fashion? From the metabolites, could the authors identify distinct groups? I am wondering if using the data to find distinct groups could also be performed (again beyond the scope of the study, but could be interesting to look at to see if there may be groups that are not clearly seen). For example, using Dirichlet Multinomial Mixtures to identify distinct groups. This could be added as a figure in the supplement. Part of me wonders if differences in serum metabolites may be associated with clinical outcomes. 

Yes, we have performed unsupervised analysis in the form of Principal Component Analysis. The results are shown in S2 Fig (line 595). However, unsupervised PCA did not show separation between the pathogen groups. We have looked into Dirichlet Multinomial Mixtures after the reviewer's suggestion, but feel that this would not be appropriate for analysis of continuous metabolite levels present in our dataset, and is primarily of relevance for microbial metagenomics studies that involved discrete observations.

11. Discussion: Page 20, Line … “Targeted …” I appreciate that the authors point out that it is difficult based on the host-metabolomic profile to tell the difference between the various pneumonias. What isn’t clear to me is why would atypical infections, in particular have such distinct host-metabolomic profile? The authors do a commendable effort into searching for metabolites which can discriminate between infections, but what is so particular that the infections create a unique host response (e.g., such as the intra-cellular nature of some of these infections Mycoplasma and Legionella).

Although our approach primarily was focused on assessing whether metabolomics profiling could be helpful in guiding empirical antimicrobial treatment of CAP, we agree that the distinct group of atypical pathogens requires further elaboration. Therefore, we added to the manuscript (355-360):

“Ideally, we would have further investigated differences within studied groups, e.g. to identify metabolic responses to specific pathogens within the atypical pathogens and viral infection groups. For example, it would be of interest to study Legionella species more in-depth because their intracellular growth might result in a differentiated host-response. However, this was considered not feasible in this study due to sample size restrictions.”

12. Discussion: Page 21, Line … “Lactic acid …” I think this is interesting because there are R and L enantiomers that are involved in microbial metabolism, but from a clinical point of view, lactemia in the serum is sign of severe disease. Perhaps, it may actually reflect severity of disease.

The reviewer raises an interesting point here. Unfortunately with the metabolomics method used, we are not able to differentiate between R and L enantiomers. Aside from that, we have added a clarification on the interpretation of the finding of lactic acid as metabolite of interest (lines 325-331):

“Lactic acid levels are also known to rise in case of severe disease. However, because the three pathogen groups were well balanced in terms of disease severity and, for example, did not show significant differences in pH levels, we hypothesize that the differences in lactate levels are, in this case, an effect of the pathogen-specific host-response to infection. The result showed that models including disease severity covariates do not perform better than models without these confounders, thus supporting this hypothesis.”

13. Discussion: Page 22, Line … “In this study, we included patients …” It’s interesting that the authors utilized a pneumonia score, perhaps to understand some of the granularity of the data the authors should try to expand the PSI score and reassess their model based on the severity of disease? Moreover, have the authors tried to separate out the analysis based upon severity? The severity of disease could serve as a confounder in their analysis. I recommend the authors split the PSI score and attempt to construct their models utilizing

We would like to refer to point 1 in response to reviewer 2, where we explain how we incorporated disease severity variables in the revised manuscript.

---

## [Decision Letter · Decision Letter 1]

14 Apr 2021

PONE-D-21-00435R1

Metabolomic profiling of microbial disease etiology in community-acquired pneumonia

PLOS ONE

Dear Dr. den Hartog,

Thank you for submitting your manuscript to PLOS ONE. After careful consideration, we feel that it has merit but does not fully meet PLOS ONE’s publication criteria as it currently stands. Therefore, we invite you to submit a revised version of the manuscript that addresses the points raised during the review process.

We look forward to receiving your revised manuscript.

Kind regards,

Aran Singanayagam

Academic Editor

PLOS ONE

Journal Requirements:

Reviewers' comments:

Reviewer's Responses to Questions

**Comments to the Author**

1. If the authors have adequately addressed your comments raised in a previous round of review and you feel that this manuscript is now acceptable for publication, you may indicate that here to bypass the “Comments to the Author” section, enter your conflict of interest statement in the “Confidential to Editor” section, and submit your "Accept" recommendation.

Reviewer #1: All comments have been addressed

Reviewer #2: All comments have been addressed

2. Is the manuscript technically sound, and do the data support the conclusions?

Reviewer #1: Yes

Reviewer #2: Yes

3. Has the statistical analysis been performed appropriately and rigorously? 

Reviewer #1: Yes

Reviewer #2: Yes

4. Have the authors made all data underlying the findings in their manuscript fully available?

Reviewer #1: Yes

Reviewer #2: Yes

5. Is the manuscript presented in an intelligible fashion and written in standard English?

Reviewer #1: Yes

Reviewer #2: Yes

6. Review Comments to the Author

Reviewer #1: I only have one suggestion:

As for 1.) no standardization for sampling times and conditions was applied.

Please add in the disscussion if this may limit some (say which ones!) of the conclusions of the study and why.

Reviewer #2: The authors have addressed all comments. They do a commendable job and also addressed missing clinical variables in their response.

7. PLOS authors have the option to publish the peer review history of their article (what does this mean?). If published, this will include your full peer review and any attached files.

Reviewer #1: No

Reviewer #2: No

---

## [Author Response · Author response to Decision Letter 1]

29 Apr 2021

Dear Dr. Singanayagam, 

Hereby we would like to resubmit our revised manuscript entitled “Metabolomic profiling of microbial disease etiology in community-acquired pneumonia.” 

We have addressed the remaining reviewer comments in this revised version, for which we provide a specific response below. In addition, we have made a number of minor textual changes to further improve readability of the Discussion section.

We hope that this manuscript is now acceptable for publication.

Sincerely, 

on behalf of all co-authors,

Ilona den Hartog and Coen van Hasselt

Reviewer #1:

I probably missed this point. When were the serum samples taken? In the morning before food? Has this been standardized? 

As for 1.) no standardization for sampling times and conditions was applied. Please add in the disscussion if this may limit some (say which ones!) of the conclusions of the study and why.

Response:

Since the samples were not collected specifically for metabolomics analysis, no standardization for sampling times and conditions was applied.

In the Methods section we stated the following (lines 85-86): 

“The samples were taken from CAP patients within 24 hours after hospital admission.” 

Furthermore, we have added to the discussion the following (lines 348 - 356):

“Furthermore, no standardization of sampling times and conditions was applied, e.g., patients had not fasted before blood sampling, which may influence the metabolite patterns found. Since variations in sampling conditions were unknown, we were unable to consider these in our analyses. However, we expect that the impact of not standardizing and correcting for these factors is limited because the noise in metabolite levels introduced by these factors is expected to be random with regard to the pathogen groups compared in this study. A standardized sampling approach could improve the sensitivity of the models to detect predictive metabolites because some noise is reduced. However, the specificity of the models with respect to the prediction of specific pathogens would be unchanged, since no correlation with pathogen groups is likely.”

---

## [Editor Report · Decision Letter 2]

17 May 2021

Metabolomic profiling of microbial disease etiology in community-acquired pneumonia

PONE-D-21-00435R2

Dear Dr. den Hartog,

We’re pleased to inform you that your manuscript has been judged scientifically suitable for publication and will be formally accepted for publication once it meets all outstanding technical requirements.

Kind regards,

Aran Singanayagam

Academic Editor

PLOS ONE
---

## [Editor Report · Acceptance letter]

25 May 2021

PONE-D-21-00435R2 

Metabolomic profiling of microbial disease etiology in  community-acquired pneumonia 

Dear Dr. van Hasselt:

I'm pleased to inform you that your manuscript has been deemed suitable for publication in PLOS ONE. Congratulations! Your manuscript is now with our production department. 

Kind regards, 

on behalf of

Dr. Aran Singanayagam 

Academic Editor

PLOS ONE